# Advantages and Limitations of the Body Mass Index (BMI) to Assess Adult Obesity

**DOI:** 10.3390/ijerph21060757

**Published:** 2024-06-10

**Authors:** Yilun Wu, Dan Li, Sten H. Vermund

**Affiliations:** 1Department of Biostatistics, Yale School of Public Health, New Haven, CT 06510, USA; lilo.wu@yale.edu; 2Department of Epidemiology of Microbial Diseases, Yale School of Public Health, New Haven, CT 06510, USA; dan.li@yale.edu

**Keywords:** body mass index, obesity, overweight, anthropometry, nutrition, adiposity, exercise, screening guidelines

## Abstract

Obesity reflects excessive fat deposits. At-risk individuals are guided by healthcare professionals to eat fewer calories and exercise more, often using body mass index (BMI; weight/height^2^) thresholds for screening and to guide progress and prognosis. By conducting a mini-narrative review of original articles, websites, editorials, commentaries, and guidelines, we sought to place BMI in the context of its appropriate use in population health, clinical screening, and monitoring in clinical care. The review covers studies and publications through 2023, encompassing consensus reviews and relevant literature. Recent consensus reviews suggest that BMI is a valuable tool for population surveys and primary healthcare screening but has limitations in predicting the risk of chronic diseases and assessing excess fat. BMI can guide nutritional and exercise counseling, even if it is inadequate for reliable individual risk prediction. BMI cut-offs must be reconsidered in populations of varying body build, age, and/or ethnicity. Since BMI-diagnosed overweight persons are sometimes physically and physiologically fit by other indicators, persons who are overweight on BMI should be more fully evaluated, diagnosed, and monitored with combined anthropometric and performance metrics to better clarify risks. The use of combined anthropometric and performance metrics involves integrating measurements of body composition with assessments of physical function and fitness to provide a more comprehensive evaluation of an individual’s health and fitness status. Eligibility for bariatric surgery or semaglutide satiety/appetite-reduction medications should not be determined by BMI alone. Awareness of the advantages and limitations of using BMI as a tool to assess adult obesity can maximize its appropriate use in the context of population health and in rapid clinical screening and evaluation.

## 1. Introduction

Body mass index (BMI) is used to screen for macro-nutritional status, whether undernutrition, normal range, or overnutrition [1]. An individual’s weight in kilograms divided by height in meters squared was devised by Adolphe Quetelet, a Belgian mathematician, in the 1830s to enable the comparison of White middle-aged European men of varying heights via a common parameter. This Quetelet index was termed “the body mass index” by Ancel Keys in 1972 [1]. Four categories of BMI are in common use today to estimate patient conditions: (1) underweight or undernourished status (<18.5 kg/m^2^); (2) normal BMI (18.5–<25.0 kg/m^2^); (3) overweight status (25.0–<30.0 kg/m^2^); and (4) persons with obesity as ≥30.0 kg/m^2^ [2]. Obesity status can be subcategorized further to capture more or less severe BMI variance [3]. BMI is relatively simple to measure in adults [4,5,6,7,8,9] and functionally sums a variety of complex body indicators. (Interpreting BMI in children is more complex and is not reviewed here.) Limitations are notable, as summarized by Bray [10]: BMI does not measure body fat distribution (e.g., it does not recognize more harmful abdominal fat), is a limited measure of body fat (inviting misclassification of persons who are muscular or have larger bone structures) [11], and does not inform practitioners of either the origin or the heterogeneity of obesity and its outcomes in specific individuals or populations. BMI has substantial utility in population-level statistics, summarizing current burdens of under- or over-nutrition and assessing trends in communities. As a simple, inexpensive clinical screening tool, BMI remains popular in patient medical encounters, though its limitations for individual patient forecasting are well documented [12]. 

Healthcare providers and public health agencies alike recognize obesity to be a risk factor for multiple outcomes such as cardiovascular disease, diabetes, chronic orthopedic problems, accidents, collagen vascular diseases, lower quality of life, osteoarthritis, and autoimmune disorders [13,14]. High BMI is also associated with an increased risk of colorectal cancer [15], primary liver cancer, and cancer mortality [16]. Obesity is a risk factor for death from SARS-CoV-2 infection (COVID-19 disease), perhaps related to stress on lung capacity and complexities in ventilator management in intensive care settings [17]. Besides compromising physical health, obesity very often leads to psychological and social distress. Overweight status is associated with an increased risk of depressive disorder among adults [18]. Individuals with excess weight are subject to social devaluation and discrimination, termed “fat shaming” by some. Such negative stereotypes because of body weight, better termed “weight stigma”, affect people with obesity; persons feeling ashamed or embarrassed may hesitate to seek timely professional help, may have trouble finding needed social support, and can impede weight loss maintenance [19]. 

The etiology of obesity has been studied widely, including genetic and environmental factors. Both obesity and BMI have been considered heritable traits. However, the genetics of obesity are not sufficient to account for the variability of BMI [20,21]. Higher BMI is influenced heavily by sedentary lifestyles, structural elements (e.g., parks, sidewalks, bicycle lanes) in the lived environment, affordability and availability of healthy foods (so-called “food deserts” are neighborhoods where fresh vegetables and fruits are not conveniently obtained), and marketing and promotion of unhealthy foods in society [22].

While acknowledging its utility as a simple health indicator useful for screening, the literature highlights how BMI can mislead healthcare providers. BMI estimates adiposity crudely, given that muscles and bones have a higher density than fat (adipose tissue), and some fat dispositions are worse than others, as with abdominal adiposity. BMI does not measure functional health and fitness but is merely an anthropometric measure. The American Medical Association (AMA) issued a report and policy recommendation in June 2023 that urged medical professionals to do more comprehensive assessments of overweight and rely less on BMI [23,24]. In their 2023 report, The Lancet Diabetes and Endocrinology Commission highlighted similar caveats in the context of the definition and diagnosis of obesity [25].

BMI was developed originally from a single sample of White, middle-aged European men [3]. Thus, the validity of BMI for the global south is compromised. Even in the global north, population diversity is increasingly the norm; in the United States, people of color made up 40% and women were 50.5% of its population in 2021 [12]. The importance of this diversity is illustrated by the South Asian body habitus that is different enough from others to suggest a need for completely different cut-offs for obesity in BMI screening [26]. Polynesian body habitus may also differ, such that different cut-offs are appropriate. In the context of these complexities, this mini-narrative review summarizes critical assessments of BMI as a primary standard to assess weight abnormalities, presents advantages and limitations of BMI in its current applications among adults, suggests how BMI can be used to screen for an individual’s weight, risk prediction for certain diseases, macro-nutritional status, and reviews complementary measures for clinicians and patients to achieve a more comprehensive understanding of current status, trends, and prognosis for overweight conditions.

## 2. Materials and Methods

The methodology employed in this study is a mini-narrative review, rather than a formal scoping review. We used the Google Scholar search engine to find popular scientific articles and news that sought to critique BMI as a health standard and/or highlighted potential bias in relying on BMI to assess weight and nutritional status. To summarize principal scientific peer views as to how BMI can be helpful or misleading, we searched sources from PubMed. In addition, we obtained alternative or complementary measures from the U.S. Centers of Disease Control and Prevention (CDC), the National Institutes of Health (NIH), The Lancet Diabetes and Endocrinology Commission, the AMA, and the World Health Organization (WHO). Finally, we assessed the arguments presented in the AMA Council on Science and Public Health critique of BMI.

## 3. Results

### 3.1. Critical Assessment

Even critical assessments acknowledge the perceived advantages of BMI, namely its simplicity and rapidity of measurement in persons able to stand up straight for height measurement and to step unclothed (or nearly so) on an accurate weight scale. Its numerical value comes at very little expense to the patient or the healthcare system; even the time and effort of healthcare workers can be minimized since community workers can be trained to obtain accurate height and weight in both higher- and lower-income regions. BMI does not involve radiation and/or imaging, technically complex anthropometric assessments, or appointments beyond the screening or primary care visit to undergo clinical procedures or lab tests [27]. Persons pursuing a weight loss plan through diet and exercise can use BMI as the anchor assessment and calculate weight goals with assistance from a primary care provider, enabling them to simply track weight over time since height in adults is stable over the short term [27]. The convenience and simplicity of BMI measurement enable clinicians and epidemiologists to estimate individuals and populations alike as to health and mortality risks and trends [27,28]. Moreover, BMI serves as a useful tool to predict the risk and prognosis of certain chronic diseases. In 15,828 patients with stable coronary heart disease, there was a progressive rise in cardiometabolic and inflammatory risk factors (e.g., hypertension, diabetes, triglycerides, and inflammatory biomarkers) at BMI measures exceeding 25 kg/m^2^. A BMI below 20 kg/m^2^ and a BMI of 35 kg/m^2^ or higher were significant indicators of a poor prognosis [29]. Nevertheless, the lowest rates of all-cause and cardiovascular mortality among these patients with coronary heart disease were observed in the BMI range of 25 to 35 kg/m^2^, suggesting that BMI alone is too simplistic to give a comprehensive risk assessment in a clinical setting without consideration of many other factors.

Among adults, BMI has been used to assess overweight status independent of age or sex [28], though its cut-offs for the elderly are debated [30]. BMI cut-offs may not hold the same predictive value for individuals over the age of 65 as they do for younger adults, as modest overweight status may not carry the same health implications for the older population [31]. This notion underscores the complexity of assessing weight status in seniors and highlights the need for nuanced interpretations of BMI measurements in this demographic. Considering additional factors beyond numerical thresholds is crucial for accurately evaluating health risks and guiding healthcare interventions in older adults [31]. BMI and z-scores in children are more complex and are not reviewed here [4,5,6,7,8,9].

One may acknowledge the advantages of BMI as a screening tool, but at least three limitations in properly assessing weight status with BMI are noted by nutritionists, clinicians, and epidemiologists. We review them briefly.

#### 3.1.1. Higher BMI Does Not Necessarily Indicate Excessive Body Fat

An “abnormal” BMI only suggests whether a measured weight (in kg) divided by height squared (in m) is out of range of so-called normal limits [27]. If a subject is a muscular athlete, for instance, a high BMI can mislead. In a sample of men with a BMI of 27 kg/m^2^ (characterized by BMI as overweight), the 95% confidence interval for body fat percentage varied from 10% to 32% [32]. Hence, the “overweight” range included men with a low estimated body fat as well as men whose body fat percentage is compatible with obesity [33]. This can apply to others with varying body habitus.

Incorporating body composition analysis techniques such as dual-energy X-ray absorptiometry (DEXA) or bioelectrical impedance analysis (BIA) alongside BMI measurements can provide a more comprehensive assessment of body fat percentage and distribution. This offers a clearer picture of an individual’s health status and is advisable for clinical assessments [34]. 

Assessing progress using BMI can be challenging for adults engaged in a fitness program that includes exercise and a healthier, calorie-controlled diet. This challenge arises because the individual may gain muscle mass while losing fat, resulting in an increased or stable BMI, which confuses the evaluation by health/fitness professionals monitoring patient/client progress. Therefore, it is crucial to recognize that relying solely on BMI may lead to a misleading assessment of progress in some cases [27].

#### 3.1.2. Simplicity of BMI Does Not Ensure Its Validity for Screening in All Populations

Although BMI is an easy measure, its corresponding categories are more complex. Cutoff values for categories have fluctuated over the past century, depending upon the given year and guideline document (Table 1). In 1993, the NIH considered men whose BMI were 27.8 kg/m^2^ or greater and women whose BMI were 27.3 kg/m^2^ or greater as being overweight [3]. In contrast, the WHO classified people with a BMI between 20 and 24.9 kg/m^2^ as having normal weight, those with a BMI between 25 and 29.9 kg/m^2^ as overweight, and those with a BMI greater than 30 kg/m^2^ as obese [3]. In 1997, the International Obesity Task Force further expanded the BMI categories into different levels of obesity: 30–34.9 kg/m^2^ (class I obesity), 35–39.9 kg/m^2^ (class II obesity), and 40 kg/m^2^ or greater (class III obesity) [3]. One year later, this Task Force modified the cutoff value between normal and overweight to 25 kg/m^2^ and did not subdivide the obesity classifications [35]. 

In 2002, guidelines for Chinese adults used yet another threshold for being overweight (BMI > 24 kg/m^2^) to reflect a study of BMI predictive value for all-cause mortality in China [36]. In 2015, in India, a BMI > 23 kg/m^2^ and <25 kg/m^2^ were regarded as overweight and >25 kg/m^2^ as obese for Indians [37]. The so-called South Asian body habitus is thought to have higher health risks at a lower BMI than among Caucasians due to a comparatively higher proportion of body fat at the same BMI [38]. Consequently, persons with more body fat at a lower BMI than considered obese by WHO cut-offs, termed “sarcopenic obesity” [30], are at higher risk of type 2 diabetes with a relative excess adiposity, but would not be successfully identified if screened by WHO BMI criteria. The BMI cutoff of age- and sex-adjusted incidence for type 2 diabetes that is equivalent to the cutoff of 25 kg/m^2^ in the White population is as low as 19.2 kg/m^2^ in South Asians [39]. Young Indians living in India were found to have more visceral and abdominal fat and reduced glucose disposal rates than Caucasians with matched age and BMI [40]. Similar work is being carried out on Polynesian populations. Variability in the prognostic value of BMI among a variety of sub-populations is very likely given the simplicity of the metric. However, even with different cut-offs, changes over time may be valid for tracking changes within individuals and summarizing population trends. Varying BMI guidelines and thresholds suggest that BMI is not a robust universal metric unless tailored for specific populations (Table 1). 

Exploring alternative metrics such as waist-to-hip ratio (WHR) or waist-to-height ratio (WHtR) in addition to BMI could offer a more comprehensive assessment of body composition and associated health risks in diverse populations. These measures specifically target central adiposity and can provide valuable insights into metabolic health that are not captured by BMI alone. By considering multiple metrics, healthcare practitioners can personalize interventions and risk assessments to better meet individual needs, improving the effectiveness of preventive strategies and treatment plans [41]. Genetic factors play a significant role in obesity predisposition and response to weight management. Variations in genes related to metabolism, appetite regulation, and fat storage influence obesity susceptibility [42]. Some speculate that understanding genetic influences can help tailor personalized risk assessments and treatment strategies, leading to more effective interventions. Current research seeks to integrate genetic risk factors into obesity assessment and management to see if precision and efficacy in interventions can be improved. Ongoing research in genetics and obesity offers promise for innovative therapeutic approaches targeting genetic factors contributing to obesity and its effects [43,44,45]. 

**Table 1 ijerph-21-00757-t001:** Examples of body mass index cutoff values from different institutions and/or for different populations.

Target Population	Underweight	Normal	Overweight	Obesity
Universal (1993 by NIH) [3]	*	*	≥27.8 for men ≥27.3 for women	*
Universal (1993 by WHO) [3]	*	20–24.9	25–29.9	≥30
Chinese adults in China (2002) [36]	≤18.5	18.6–23.9	24–27.9	≥28
Indians in India (2015) [37]	*	*	23–24.9	≥25
South Asians (2023) [43,46]	≤18.5	18.6–22.9	≥23	*

* Threshold not indicated. NIH = U.S. National Institutes of Health; WHO = World Health Organization.

#### 3.1.3. BMI Is Not an Adequate or Comprehensive Predictor of Higher Disease Risk

A study of 5344 participants aged >55 years old suggested that the benefits of physical activity may compensate to some extent for the adverse predictions of a higher BMI [44]. Another study of 899 women showed that lean but unfit women had a higher risk of all-cause mortality than overweight but more fit women, i.e., BMI was less predictive than good fitness [45]. Termed by some observers as the “fat but fit” paradigm or an “obesity paradox”, better mortality outcomes in persons with overweight or obesity may occur in certain race-ethnicity-sex-age strata [47,48], and fitness level must be considered while assessing the association between body weight and medical outcomes [49]. As mentioned previously, BMI estimates fat amount but is not precise in measuring fat distribution, as with abdominal fat or central fatness, which are better predictors of the risk of mortality [50]. 

Standard BMI categories can lead to misclassification bias when considering age, ethnicity, race, sex, or metabolic disease risks for certain disorders [51]. In a study of 40,420 persons aged >18 years in the US, researchers used nutritional status measures, including cholesterol and glucose, to determine whether subjects were at risk of cardiometabolic health disease by comparing their predictive values with BMI; modeled results suggested that >70 million US adults would be misclassified as cardiometabolically healthy or unhealthy by BMI [52]. Among US adults aged >50 years, a multilocus genetic risk score for BMI was estimated for participants born between 1900 and 1958; an interaction of birth cohort and known genetic variants with BMI suggested that genetic risks might be exacerbated by obesogenic exposures [53]. Apart from genetic factors, the relationship between BMI and percent body fat is highly variable by race, ethnicity, sex, and age (acknowledging that these simple constructs may mask more complex social elements), and these factors may interact [21]. For example, for the same BMI, females have a higher percent of fat than males, and for similar fat-free body mass, Black women had significantly more percent of fat and, hence, a higher body weight than White women [54]. Incorporating advanced imaging techniques such as dual-energy X-ray absorptiometry (DEXA) or magnetic resonance imaging (MRI) could provide a more accurate assessment of body composition, including fat distribution and muscle mass. These methods provide greater precision in identifying the risk of metabolic diseases and can help overcome the limitations of BMI in specific population groups. By integrating multiple assessment modalities, healthcare practitioners can gain a more comprehensive understanding of an individual’s health status, enabling more personalized interventions and better outcomes. 

### 3.2. Comprehensive Macro-Nutritional Assessments

BMI will continue to be a simple and convenient tool for assessing macro-nutrition at the population level, recognizing, of course, that BMI does not measure micronutrients [55]. Despite efforts by the WHO and other agencies, BMI thresholds continue to be debated, and we now appreciate that race, ethnicity, sex, age, and fitness must be considered as covariates. Basing an obesity diagnosis solely on BMI can be inaccurate and may incur discriminatory and costly patient consequences, as when employers increase employees’ health insurance costs if the BMI is above a given threshold [52]. In healthcare systems, bariatric surgery may be restricted to patients with a high BMI (e.g., >35 kg/m^2^); a more objective assessment (co-incident type 2 diabetes, for example) could address the limitations of using BMI eligibility criteria alone [12]. 

Following community-based or primary care screening, clinical referral of overweight persons to qualified nutrition and healthcare professionals can solicit a more detailed body composition and nutritional status evaluation. Many alternative ways to assess body habitus are used by clinical nutrition experts for more detailed and nuanced assessments. Waist circumferences can assess body fat distribution by measuring the distance around the abdomen in different anatomical locations [56]. One disadvantage of using waist circumference to assess body fat distribution is that it provides a general measure of abdominal fat without distinguishing between subcutaneous fat and visceral fat. The waist-to-hip ratio uses waist and hip circumference measurements and may be a better predictor of cardiovascular disease occurrence than BMI [13]. The waist-to-hip ratio focuses specifically on the ratio between waist and hip measurements, potentially overlooking other important factors that contribute to overall body composition and health status. The waist-to-height ratio was found to be associated with dyslipidemia and hyperuricemia in a sample of 4565 adults in China [57]. One key limitation of waist-to-height is that it does not account for individual variations in body shape, muscle mass, and bone structure. These factors can influence waist circumference measurements independently of actual fat distribution [58]. These measures are also relatively simple in the hands of trained practitioners and are low-cost; they can better evaluate adiposity instead of an exclusive focus on excess weight. 

Semaglutide drugs (e.g., Ozempic^®^ or Wegovy^®^) provide glucagon-like peptide-1 (GLP-1) receptor agonists in once-weekly injections. Work-to-date suggests that semaglutide mimics the GLP-1 hormone released in the gastrointestinal tract with food challenges, stimulating more insulin production, reducing blood glucose, and promoting satiety sensations and appetite suppression [59]. Semaglutide injections may not be covered by health insurance for people falling below certain BMI cut-offs; a more comprehensive assessment could provide a more accurate assessment of need [60].

Skinfold thickness is considered a valuable tool but requires considerable technical skill and experience to be measured accurately. The assessment of total body fat requires the integration of skinfold thickness measurements at different body sites. The integration of data considers the number of sites and their locations, with sex-specific adjustments [61]. 

Multiple models have been constructed to analyze body compositions that are more complex than fat mass and fat-free mass alone, incorporating classifications beyond adipose and lean tissue. In a two-component model, bioelectrical impedance analysis (BIA) is used to acquire total body water by measuring the resistance to a small electrical current that travels between skin electrodes. Multifrequency BIA and bioimpedance spectroscopy (BIS) both distinguish between intracellular and extracellular water to detect fluid shifts and evaluate hydration levels [62]. BIS in HIV-infected breastfeeding South African mothers provided valid measures of fat-free mass, whereas BMI was helpful only in predicting fat mass [63]. Uncommonly used in general medicine, a four-compartment model using dual X-ray absorptiometry (DXA) and air displacement plethysmography (ADP) measures the mineral content in the bone, while body volume and dilution techniques assess total body water [64]. An automated computer vision method estimates total body fat via visual body composition (VBC) that uses two-dimensional photographs from a conventional smartphone camera and convolutional neural network algorithms; validations were promising versus more burdensome DXA, BIA, and ADP measures [65].

Costly strategies to reliably quantify body composition at the tissue–organ level include cross-sectional imaging with whole-body magnetic resonance imaging (MRI) or computed tomography (CT). Both can differentiate between organs and muscles, the two subcomponents of fat-free mass. Though cross-sectional imaging can illustrate longitudinal changes in a single individual’s body composition, it is harder to interpret imaging for comparative predictions for groups of people [66]. Radiation exposure in both DXA and CT is a disadvantage. Although MRI does not involve ionizing radiation, it requires one to be stable in a loud and noisy environment for a long time, which can be costly, tiring, and disruptive for patients, particularly those with claustrophobia. Three-dimensional photonic scanning (3DPS), positron emission tomography (PET), and single-photon emission computed tomography (SPECT) are comparatively costly and complex imaging tools that assess adiposity and some physiological functions [67]. Healthcare inequities are evident when one considers the cost and availability of advanced diagnostics in lower-resource settings.

In light of these advancements and challenges, recommendations have been proposed to enhance the utilization of anthropometry in nutrition assessment and clinical care. These recommendations include reconsidering BMI cut-offs to account for demographic variations, conducting comprehensive nutritional and anthropometric examinations, incorporating performance metrics to supplement BMI assessments, and avoiding sole reliance on BMI for determining eligibility for interventions. Implementing these recommendations can significantly improve the effectiveness of anthropometry in assessing population health and guiding clinical care.

## 4. Discussion

The BMI is helpful when it is used as a screening test and population-appropriate cut-offs are used. More extreme BMI findings have a high predictive utility for obesity-related medical conditions, as well as for undernutrition and cachexia. The obesity crisis or progress in reducing undernutrition cannot be assessed by BMI alone; however, the global rise in obesity and decline in underweight are attributed to a shift in the population distribution of BMI, and this can mask social inequity both within and across populations [63]. 

BMI provides valuable information at a population and community level, particularly for assessing trends. However, BMI must not be confused with an authoritative standard for assessing adult body habitus, given its limitations in estimating body fat amount or adipose tissue distribution. The 2023 AMA report states: “There are issues with BMI in diagnosing obesity. Further measures are recommended for clinical use. BMI loses its predictability in assessing the fat mass on the individual level because of the body variation among different age, sex, and race groups. Therefore, BMI should not be used as the only criterion in decision- or policy-making” [18]. Advances in data sciences and artificial intelligence may help address guideline discrepancies by refining criteria and cut-offs by key relevant cofactors like race, ethnicity, sex, age, and fitness. 

There are several variables that may be able to address BMI’s limitations. Body Fat Percentage measures the proportion of fat relative to total body weight, providing a more precise assessment of body composition than BMI. This metric can be evaluated through skinfold measurements, bioelectrical impedance analysis, DEXA scans, and hydrostatic weighing [68]. The waist-to-hip ratio (WHR), determined using a simple tape measure, indicates fat distribution and serves as a reliable predictor of cardiovascular risk. Similarly, waist circumference, also measured with a tape measure, assesses abdominal fat, with elevated values associated with a higher risk of metabolic diseases [69]. Lean body mass, essential for evaluating muscle health and overall metabolic function, measures the body’s mass excluding fat and can be assessed through DEXA scans, hydrostatic weighing, and bioelectrical impedance analysis [70]. Cardiorespiratory fitness [71] (CRF) assesses the efficiency of the cardiovascular and respiratory systems in supplying oxygen during sustained physical activity and is a robust predictor of overall health and mortality.

## 5. Conclusions

Nutritional epidemiology faces continuing challenges in assigning BMI a suitable screening and preliminary evaluation role; further evidence-based consensus is needed for guidelines and diagnostic thresholds in varying populations. Medical evaluations with greater predictive value need accurate measures and a definitive diagnosis of harmful adiposity, as well as functional assessments; such assessments encounter health system constraints of accessibility and affordability [64]. Simple, point-of-care assessment tools to better assess macro-nutritional status could supplant BMI as a screening tool of choice, but these are not yet affordable and/or validated.

The potential innovative approach could involve utilizing advanced technologies such as bioelectrical impedance analysis (BIA), dual X-ray absorptiometry (DXA), or air displacement plethysmography (ADP) for body composition analysis. These methods provide more detailed information about body fat distribution and composition compared to BMI, which only considers weight and height. Additionally, incorporating blood variables such as lipid profile, glucose levels, and inflammatory markers can offer insights into metabolic health and obesity-related risks beyond what BMI alone can provide [62]. A vital research agenda is the comparison of BMI screening test efficiencies (sensitivities and specificities) and population-specific positive and negative predictive values. Such work may improve the specific combinations of screening, diagnostic, and predictive tools, with BMI as a likely component of efficient algorithms. Workforce issues must be considered in such optimization efforts, as the need for skilled practitioners, test risks, and costs are all factors that determine the extent to which more complex approaches are useful in specific settings. 

By combining advanced body composition analysis techniques with comprehensive metabolic and fitness testing, healthcare professionals can gain a more comprehensive understanding of an individual’s health status and obesity-related risks. This personalized approach to obesity assessment considers factors such as muscle mass, visceral fat levels, and metabolic health markers, in addition to weight status [64]. Leveraging data science and artificial intelligence to analyze the complex interactions between body composition metrics and blood-derived variables can enhance the accuracy and predictive power of this innovative approach. Developing algorithms that integrate multiple data points can help healthcare providers tailor interventions and treatment plans more effectively to address obesity and its associated health risks [66]. Fitness must be assessed alongside other indicators [72,73].

Combining advanced body composition analysis with comprehensive blood variables offers a promising strategy for improving obesity assessment at the population level and advancing personalized healthcare approaches to combat obesity and its related health consequences [61,67]. While BMI remains a cornerstone of public health surveillance and screening efforts, its limitations necessitate the embracing of emerging technologies and adopting a multifaceted approach to body composition assessment to better address the diverse needs of populations worldwide. 

## Data Availability

The first author can share information not contained in the manuscript: lilo.wu@yale.edu.

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
