# Peer review of "Advantages and Limitations of the Body Mass Index (BMI) to Assess Adult Obesity"

_ijerph, 2024, doi:10.3390/ijerph21060757_

Round 1
Reviewer 1 Report
Comments and Suggestions for Authors
Thank you for the opportunity to review the article named “Advantages and Limitations of Body Mass Index (BMI) to Assess Adult Obesity”
This is a very interesting review which gives the reader the opportunity to analyze and reconsider the usefulness of this marker in daily clinical use.
However, there are some aspects to consider.
Introduction
Table 1 is not a table
I consider that the introduction should focus more on the negative aspects of obesity and its causes, also referring to the genetic and ethnic factor which influences the BMI to be questioned, and the markers that are used (not only the BMI) and its advantages and disadvantages in brief, at the end of the introduction
Methodology
I consider it important to show in a flow chart the methodology with which the analyzed articles were chosen or rejected.
The methodology mentions the use of BMI as a marker of obesity and nutritional status, they are two factors that, despite having common markers, are determined genetically, and factorially in many aspects independently, I believe the analysis should be separate.
If the prism methodology was not used, what type of methodology was used?
Results
In the first paragraph of the results mentioning the advantages of using the BMI, studies should be mentioned in which this measurement has served as a compromised factor to prevent chronic diseases such as metabolic syndrome, different types of cancer, etc. in different populations.
It would be very interesting and necessary in the following paragraphs in which the disadvantages of BMI measurement are mentioned, the way in which it could be complemented or replaced, otherwise it seems only a criticism and not a scientifically based reflection.
In paragraph 3.1.2, it would be interesting to mention the genetic dimension of obesity and its complications.
In paragraph 3.2 Many topics are referred to in a somewhat chaotic manner, for example the paragraph entitled nutrition assessment and it reference about semaglutide for the management of diabetes. This comment should be in another paragraph.
Reference should also be made to the biochemical markers that could be used to evaluate the nutritional status
Author Response
Introduction
Table 1 is not a table
Response: We have removed Table 1 and include the CDC website as a reference in making the point to recognize obesity as a risk factor for multiple health outcomes. We have changed the order of Table 2 to Table 1 accordingly.
I consider that the introduction should focus more on the negative aspects of obesity and its causes, also referring to the genetic and ethnic factor which influences the BMI to be questioned, and the markers that are used (not only the BMI) and its advantages and disadvantages in brief, at the end of the introduction
Response: We have taken the reviewer's advise. We have expanded the negative aspects of obesity from health outcomes to psychological and social distress. We have also mentioned the genetic factor of obesity and how its heritability cannot explain the variability of BMI, which can be affected by environmental factors as well..
Methodology
I consider it important to show in a flow chart the methodology with which the analyzed articles were chosen or rejected.
Response: We appreciate the reviewer's comments. Since we do not wish to misrepresent our paper, we have stated clearly that it is not a full-fledged scoping review. We fear that adding a PRISMA-style flow diagram indicating the numerical values of each step of the screening process will do the paper a disservice by claiming it to be more than it is. However, we have expanded the details of the methodology section.
The methodology mentions the use of BMI as a marker of obesity and nutritional status, they are two factors that, despite having common markers, are determined genetically, and factorially in many aspects independently, I believe the analysis should be separate.
Response: Thank you for the feedback. We completely agree with the reviewer’s insightful point. We have added this distinction to the main text. More specifically, 3.1 focused more on BMI as a marker of obesity, and 3.2 focused more on BMI as a guide to nutritional status.
If the prism methodology was not used, what type of methodology was used?
Response: We have expanded the details of the methodology section. We conducted a mini-narrative review to synthesize existing literature, critiques, and perspectives on the topic of BMI as a health standard and its potential biases. We used our judgment as to relevant articles, rather than a rigorous PRISMA-style scoping review methodology.
Results
In the first paragraph of the results mentioning the advantages of using the BMI, studies should be mentioned in which this measurement has served as a compromised factor to prevent chronic diseases such as metabolic syndrome, different types of cancer, etc. in different populations.
Response: We have added this information as well as citing a relevant study.
It would be very interesting and necessary in the following paragraphs in which the disadvantages of BMI measurement are mentioned, the way in which it could be complemented or replaced, otherwise it seems only a criticism and not a scientifically based reflection.
Response: Regarding the limitations of BMI measurement, we have thoroughly addressed issue in entire paragraphs of the manuscript. We present alternative and complementary approaches to BMI measurement, based on current research findings.
In paragraph 3.1.2, it would be interesting to mention the genetic dimension of obesity and its complications.
Response: We appreciate your suggestion regarding the inclusion of the genetic dimension of obesity and its complications in paragraph 3.1.2. and we now address this aspect in the revised version of the manuscript. In paragraph 3.1.2, we have incorporated a discussion on the role of genetic factors in predisposing individuals to obesity and influencing their response to weight management strategies.
In paragraph 3.2 Many topics are referred to in a somewhat chaotic manner, for example the paragraph entitled nutrition assessment and it reference about semaglutide for the management of diabetes. This comment should be in another paragraph.
Response: Thank you for your feedback on paragraph 3.2. We have revised it to improve coherence and flow to address your concerns. As the topics are now rearranged for a more organized flow of information, we believe that this enhances the clarity and readability of the manuscript.
Reference should also be made to the biochemical markers that could be used to evaluate the nutritional status
Response: Thank you very much for the feedback. Certain nutritional measures include cholesterol, triglyceride, and glucose. We have indicated these in the results section of the review.
Reviewer 2 Report
Comments and Suggestions for Authors
It is a short, easy-to-understand article that clearly explains the advantages and disadvantages of BMI
Author Response
It is a short, easy-to-understand article that clearly explains the advantages and disadvantages of BMI
Response: Thank you.
Reviewer 3 Report
Comments and Suggestions for Authors
In the attachment

Author Response
Dear authors,
Congratulations on your scientific product mini-review.
I suggest supplementing it by considering the comments, increasing the publication's scientific and practical value.
Good luck.
Abstract:
- Please provide information on the databases you have reviewed and the years covered in your research.
Response: We have expanded our methodology section to address this suggestion.
- Your study is a mini-review, so make this clear in your writing and distinguish it from a commentary or narrative review.
Response: We acknowledge your comment about the nature of our study, and we have clarified it in the abstract and throughout the paper.
- Lines 15, 17 and 19: What do you mean by 'risk of what'? Are you referring to the risk of chronic non-communicable diseases?
Response: We indeed refer to the risk of chronic diseases. We have clarified this in the manuscript to ensure clarity for the readers.
- Line 22: Please explain what you mean by 'with combined anthropometric and performance metrics'. Do you mean body composition status and some motoric tests (which)?
Response: In our abstract, we explain the relationship between combined anthropometric and performance metrics in the study in our abstract.
- It would be helpful if you could provide a summary of the aim of your study (the context of the appropriateness of using BMI that you acknowledge) at the end of the abstract.
Response: We appreciate your suggestion to provide a summary of the aim of our study at the end of the abstract. We have incorporated this summary, emphasizing the context of the appropriateness of using BMI in population health clinical settings and clinical care screening and monitoring. The Lancet commission and the U.S. AMA reports in 2023 motivated us and they are cited clearly.
Introduction:
- Lines 29-30: Please provide the reference for that claim.
Response: We have added the reference.
- Lines 38-39: Please clarify what you mean by "measuring BMi of children is more complex." Did you mean interpreting the BMI of the pediatric population rather than measuring it?
Response: We acknowledge the need for clarity in our introduction and we have reworded this.
- Line 48: Please include an additional scientific source, such as a (systematic) review. At the moment, it's only a newspaper article.
Response: We appreciate your noting our omission. We have added the reference.
- Table 1: Although the text of Table 1 is visible in the PDF version, the table itself is not visible.
Response: We have removed Table 1 in the revised manuscript and have substituted the U.S. CDC reference instead.
- Please explain the abbreviations (LDL and HDL cholesterol) in the text and below the table.
Response: We have removed Table 1 in the revised manuscript.
- Line 104: Please specify the number of websites, articles, scientific studies and (systematics/narrative) reviews among the 120 reviews.
Response: We appreciate the reviewer's comments and repeat our rationale as explained about for Reviewer 1: Since we do not wish to misrepresent our paper, we have stated clearly that it is not a full-fledged scoping review. We fear that adding a PRISMA-style flow diagram indicating the numerical values of each step of the screening process will do the paper a disservice by claiming it to be more than it is. However, we have expanded the details of the methodology section.
Results:
- Lines 117-119: An adult's height is commonly believed to remain relatively stable. However, assessing progress can be difficult when an adult engages in a fitness regime combining exercise and a healthier, calorie-controlled diet. This is because the individual may gain more muscle while losing fat, which can increase BMI and confuse the assessment of a health worker monitoring their progress. Therefore, somehow (if you agree), write that this tool (BMI) and the accompanying commonly used strategy of a health care professional (diet + exercise) should be placed against the expected result and, consequently, a false assessment of progress based only on BMI.
Response: We appreciate this insight. We have added this observation to the result section.
- Line 124: Please insert the reference number 23 before the comma.
Response: We appreciate the reviewer's comments. We have made the change accordingly for reference number 28 in the revised manuscript.
- Line 126: Please explain the claim that "weight status may not be as serious in the elderly". You probably thought that higher weight (overall higher weight, only muscle mass?) in seniors is somewhat protective against common NCDs. Rephrase it or/and add the explanation.
Response: We have rephrased the statement and added an explanation to clarify the claim of our statement. The explanation underscores the need for considerations when assessing weight status in older adults.
- Lines 126-127: You mentioned for the second time that using the BMI tool in children is »more complex«. At least write more in the introduction because it is separate from your aim in the results.
Response: We have revised the introduction.
- 3.1.1.: Supplement content related to progress that does not necessarily involve the muscular individual. For example, an overweight or obese person who adheres to the often suggested measure of "eat less and exercise more" will lose some body fat and gain muscle if not in excessive caloric restriction. Their BMI may increase if they gain more muscle, but this person will still not be muscular. So, you add to the problem of using the BMI tool to assess progress in non-muscular individuals.
Response: We have added this insight to the results section.
- Lines 159-160: Please add the reference for the claim regarding the association between sarcopenic obesity and NCDs.
Response: The reference for the claim about the link between sarcopenic obesity and NCDs has been included (reference number 29).
- Table 2: Please ensure that the dashes are the same length.
Response: We have made the change accordingly so that the dashes are the same length in Table 2 in the revised manuscript.
- Lines 173-174: The study says a little differently, i.e. not that physical activity can compensate for the prediction of a higher BMI, but that there are benefits of physical activity in terms of NCDs independent of BMI status/change. Think it over and rephrase it.
- Consider whether there is an 'obesity paradox'. I suggest reviewing additional literature, such as Bell et al. (2015), Hansen et al. (2017), Lassale et al. (2017), and NHS (EU Congress of Obesity, 2017). If you disagree, I suggest supplementing the explanation of 'fat but fit' and 'obesity paradox' more precisely. Regardless, you can use more extensive studies or even review research to support your view further.
Response: We have provided rewrites to clarify what we believe is justified by the literature, and expanding our number of references considerably.
- Line 179-180: Please explain the relationship between an individual's fitness level, body weight, and medical outcomes.
Response: We have sought to clarify this with new language.
- Lines 221-258: Please keep in mind that when discussing other methods that complement/alternate the BMI tool, it's also important to mention their disadvantages (like for the BMI tool).
Response: We address alternative methods for assessing body composition, and have including their limitations and challenges. We agree with the review that itt is important to have a comprehensive understanding of the strengths and weaknesses of each method.
Discussion:
- The discussion should be significantly more extended, and your results, i.e., the BMI tool, should be discussed in terms of other options.
Response: Thank you very much. We have discussed additional tools and options in the discussion.
- Line 226: Please insert the reference number 58 before the dot.
Response: We have made this change.
- Provide specific recommendations for improving the use of anthropometry in nutrition assessment in the context of population health screening and clinical care monitoring, as stated in the abstract.
Response: We appreciate your attention to detail. We have addressed this point in the results. We provided specific recommendations for improving the use of anthropometry in nutrition assessment within the context of population health screening and clinical care monitoring.
Conclusion:
- Make the conclusion concrete. Consider the potential benefits of a new approach to checking obesity at the population level. Instead of relying on BMI, we propose using a method of body composition in conjunction with blood variables. Although tested on a smaller yet random and representative sample, this method could yield more generalized and accurate results for the population. Moreover, it allows for comparing obesity rates across ethnic groups with minimal/lesser misclassification errors. Let's explore the potential of this innovative approach.
Response: Thank you for your feedback. We have addressed this point in the conclusion of our study. To propose a definitive method of body composition analysis in conjunction with blood variables as a new approach to checking obesity at the population level is beyond our scope, however. We are presenting alternatives, not a definitive new approach.
Recommended Reference:
- Bell, J. A., Hamer, M., Sabia, S., Singh-Manoux, A., Batty, G. D., & Kivimaki, M. (2015). The natural course of healthy obesity over 20 years. Journal of the American College of Cardiology, 65(1), 101–102. https://doi.org/10.1016/j.jacc.2014.09.077
- Hansen, L., Netterstrøm, M. K., Johansen, N. B., Rønn, P. F., Vistisen, D., Husemoen, L. L. N., Jørgensen, M. E., Rod, N. H., & Færch, K. (2017). Metabolically Healthy Obesity and Ischemic Heart Disease: A 10-Year Follow-Up of the Inter99 Study. The Journal of clinical endocrinology and metabolism, 102(6), 1934–1942. https://doi.org/10.1210/jc.2016-3346
- 24th European Congress on Obesity (ECO2017), Porto, Portugal, May 17-20, 2017: Abstracts. Obes Facts. 2017 May 19;10 Suppl 1(Suppl 1):1-274. doi: 10.1159/000468958. Epub ahead of print. PMID: 28528328; PMCID: PMC5661480
Response: Thank you for these; these are now included.